

# Strengths and weaknesses of three Machine Learning methods for pCO$_2$ interpolation

Jake Stamell[1], Rea R. Rustagi[1], Lucas Gloege[1], and Galen A. McKinley[1,2]

[1]Columbia University, New York, NY 10027 USA
[2]Lamont-Doherty Earth Observatory, Palisades, NY 10964 USA

**Correspondence:** Jake Stamell (jake.stamell@columbia.edu)

**Abstract.** Using the Large Ensemble Testbed, a collection of 100 members from four independent Earth system models, we test three general-purpose Machine Learning (ML) approaches to understand their strengths and weaknesses in statistically reconstructing full-coverage surface ocean pCO$_2$ from sparse in situ data. To apply the Testbed, we sample the full-field model pCO$_2$ as real-world pCO$_2$ collected from 1982-2016 for each ensemble member. We then use ML approaches to reconstruct

the full-field and compare with the original model full-field pCO$_2$ to assess reconstruction skill. We use feed forward neural network (NN), XGBoost (XGB), and random forest (RF) approaches to perform the reconstructions. Our baseline is the NN, since this approach has previously been shown to be a successful method for pCO$_2$ reconstruction. The XGB and RF allow us to test tree-based approaches. We perform comparisons to a test set, which consists of 20% of the real-world sampled data that are withheld from training. Statistical comparisons with this test set are equivalent to that which could be derived

using real-world data. Unique to the Testbed is that it allows for comparison to all the "unseen" points to which the ML algorithms extrapolate. When compared to the test set, XGB and RF both perform better than NN based on a suite of regression metrics. However, when compared to the unseen data, degradation of performance is large with XGB and even larger with RF. Degradation is comparatively small with NN, indicating a greater ability to generalize. Despite its larger degradation, in the final comparison to unseen data, XGB slightly outperforms NN and greatly outperforms RF, with lowest mean bias and

more consistent performance across Testbed members. All three approaches perform best in the open ocean and for seasonal variability, but performance drops off at longer time scales and in regions of low sampling, such as the Southern Ocean and coastal zones. For decadal variability, all methods overestimate the amplitude of variability and have moderate skill in reconstruction of phase. For this timescale, the greater ability of the NN to generalize allows it to slightly outperform XGB. Taking into account all comparisons, we find XGB to be best able to reconstruct surface ocean pCO$_2$ from the limited available

data.



## 1 Introduction

The ocean is integral in mitigating climate change, as it acts as a sink for 39% of industrial-era fossil fuel carbon (Friedlingstein et al., 2019). Though independent methods agree that the global mean uptake of anthropogenic carbon by the ocean was 2.0-2.6

PgC/yr for the past several decades (Friedlingstein et al., 2019; Gruber et al., 2019), there remains much to learn about spatial and temporal variations across the ocean due to biological and physical processes.

A significant challenge is estimating the net contemporary air-sea $CO_2$ exchange from sparse $pCO_2$ observations. Currently, $pCO_2$ observations are primarily sourced from volunteer observing ships traversing the ocean, and these data are limited in time and space, covering only about 2% of all $1°x1°$ gridpoints for the last several decades (Bakker et al., 2016; Sabine et al., 2013).

Various approaches have been applied to extrapolate these sparse data to global coverage (Rodenbeck et al., 2015), based on satellite observations and climatological fields of driver variables, such as sea surface temperature, salinity, mixed layer depth, and chlorophyll. These variables are linked to both the physical and chemical processes that drive carbon uptake. As $pCO_2$ observation-based products have been refined, they have converged and now provide an independent estimate of interannual to decadal variability in the global ocean carbon sink (McKinley et al., 2020; Friedlingstein et al., 2019; Gregor et al., 2019).

Despite convergence of these methods on the global scale, we wish to better assess the fidelity of the reconstructions and determine the paths forward for further improvements. This is challenging particularly because of the limited spatial and temporal availability of independent data to which to compare (Landschützer et al., 2016; Gregor et al., 2019), where independent data refers to data not used in training of the machine learning (ML) algorithms.

The self organizing map feed forward neural network (SOM-FFN) is one of the approaches that performed well in the first

intercomparison on interpolation approaches (Rodenbeck et al., 2015). This ML approach has subsequently become a well-received method for interpolating the $pCO_2$ field (Landschützer et al., 2013, 2014, 2015, 2016, 2019). SOM-FFN first splits the data into large regions and then fits a neural network given observed $pCO_2$ and driver data. Full-field driver data is then used to create an estimate of $pCO_2$ at each location for each month. Gloege et al. (2020) show, using the Large Ensemble Testbed (LET), that the SOM-FFN accurately reconstructs the seasonal cycle of $CO_2$ flux with little bias. Gloege et al. (2020) also

indicate that decadal variability is likely overestimated in the SOM-FFN, particularly in the Southern Ocean.

Gregor et al. (2019) used an ensemble of 6 supervised ML methods to create a product that performs marginally better than SOM-FFN and other observation-based $pCO_2$ products on independent datasets. The ensemble of Gregor et al. (2019) includes three commonly used ML methods: feed-forward neural networks (NN), extremely randomised trees (ERT), and gradient boosting machine (GBM). Here, we evaluate the performance of unique implementations of these three ML methods

for the task of $pCO_2$ extrapolation from sparse SOCAT data using the Large Ensemble Testbed (LET) (Gloege et al., 2020).

The LET consists of 100 members across four independent initial-condition ensemble models. Each individual member is a realistic representation of the real ocean climate system within which $pCO_2$ is known across the entire ocean at every point in time. This allows us to ask the question, "given $pCO_2$ data as sampled in the real-world, with what accuracy can we reconstruct $pCO_2$ at every point across the global ocean?" We can determine which of our three methods provides the best





reconstruction across a range of metrics. The 100 independent members in the LET provide a large state space with which we
can test reconstruction performance across an extensive range of plausible ocean climate states.

Furthermore, we use the LET to understand the strengths and weaknesses of these three machine learning approaches in
reconstructing the mean $pCO_2$ field to estimate the variability on seasonal, sub-decadal, and decadal time scales. Along with
identifying which of our specific implementations of these ML algorithms performs best, we identify the likely errors driven
by the underlying formulation of these approaches and features of the data itself that would impact other implementations.

We test NN, XGBoost (XGB) and random forest (RF) regression. Our XGB and RF are similar to the GBM and ERT of
Gregor et al. (2019), respectively. The three approaches we test have different properties, advantages, and drawbacks. General
descriptions are provided below; for an in depth description the reader is referred to Hastie et al. (2009).

1. **Feed forward neural network (NN):** The NN acts as the baseline because it is expected to perform in a similar manner
65       to the SOM-FFN. A NN sequentially applies many linear and non-linear transformations to the input data to create a
high-dimensional representation of the data. This process can be thought of as constructing regions in the feature space.
The representation is then converted to an estimate of $pCO_2$. Benefits of the NN are that it can successfully apply highly
complex and non-linear functions to minimally pre-processed data to build a robust reconstruction. A key step in building
a NN is determining its architecture. Key choices are the number of layers and number of nodes in each layer, which
70       are the primary drivers of how many parameters need to be estimated. The size of the network can be tuned for the
type of data modeling task, and larger networks could require massive amounts of data to train properly. While this
method is very powerful, it comes with several challenges. The first is minimal interpretability; that is, we will not know
why it is making certain predictions. Second, neural networks can be unstable. As one example, two NNs with identical
architectures but two different random initializations can have drastically different performance. To compensate for the
75       initialization problem, a standard approach is to train the network multiple times and select the best performer. Another
common instability arises from the fact that perturbing a set of inputs slightly can produce quite different outputs. In other
words, two data points with similar driver variable values could have considerably different $pCO_2$ estimates. Lastly, an
important consideration for NN is that its computational load is substantially higher than those of the tree-based methods.

2. **eXtreme Gradient Boost (XGB):** The XGB model is a tree-based ensemble model, and it is one implementation of a
80       GBM. In recent years, it has become very popular in industry due to strong performance across a range of tasks as well
as relatively low computational requirements (Chen and Guestrin, 2016). This approach constructs a series of shallow
decision trees, each of which individually has very poor performance. However, each successive tree is constructed to
adjust for the errors of the previous trees. This is accomplished by fitting new trees on the residuals (taking the difference
between actual and estimate) from the previous trees. To construct each tree, the data is sampled with replacement and
85       then binary decisions are applied. These binary decisions break down the feature space into regions using linear decision
boundaries, with a single prediction applied in each region. Finally, the prediction for a point is a sum across hundreds or
thousands of trees, producing a strong model by aggregating many weak estimators. The resulting predictions are highly
complex and non-linearly related to input variables. To provide a concrete example, consider the case of two features





that could be related to pCO$_2$: latitude and longitude. A decision tree would segment the world into contiguous distinct

regions based on which latitudes and longitudes are of similar pCO$_2$. Due to such a physical relationship, one of the benefits of the XGB is intelligibility. It can easily be determined which features are driving a prediction. However, this benefit of the XGB approach also drives its biggest drawback. Because it applies one estimate of pCO$_2$ in each region, it can have trouble extrapolating to new areas for which it has not "seen" data.

3. **Random forest (RF):** The RF (Breiman, 2001) is an older tree-based approach in which N decision trees are indepen-

dently built by bootstrapping the data. There is no reference to the error of the previous trees, although the trees will be correlated. The final RF prediction is an average across the predictions from all N trees and is able to capture complex non-linear decision boundaries. Though its popularity for commercial applications has been eclipsed by XGB, it remains a standard component of the ML toolkit. We include it as a point of comparison to the other approaches and as an alternative to XGB.

It is important to state the key assumptions made across all these approaches. The first assumption is to treat each data point independently. Instead of using ML algorithms that take into account the spatial and temporal aspects of the data, these approaches treat each data point separately, regardless of if they are adjacent in space or time. We use time and location as additional features in our ML algorithms. The second assumption, supported by previous studies, is that we can estimate ocean pCO$_2$ statistically by association with physical and biological variables. Our third assumption is that the real-world sampled

data cover sufficient state space of the whole ocean such that extrapolation is possible. In actuality, we have a correlated and biased sample of data because of the real-world data collection process in which observations are concentrated along major shipping routes. A commercial ship traversing the ocean collects data points along a trajectory between major shipping ports, leading to along-track correlations (Jones et al., 2012). Data are more dense in the Northern Hemisphere, while Southern Hemisphere data are very sparse (Bakker et al., 2016). Given these concerns, a benefit of the LET is that we can compare the

ML model output to the ground truth to see how well it reconstructs, despite these assumptions that are not entirely valid in the real world.

## 2 Methods

### 2.1 Data inputs

The 100-member Large Ensemble Testbed (LET) consists of 25 randomly selected members for 1982 - 2016 from each of four

independent initial-condition ensemble of Earth system models:

– CanESM2: Second Generation Canadian Earth-System Model (RCP8.5) (Fyfe et al., 2017)

– CESM-LENS: Community Earth System Model – Large Ensemble (RCP8.5) (Kay et al., 2015)

– GFDL-ESM2M: Geophysical Fluid Dynamics Laboratory Earth-System Model (RCP8.5) (Rodgers et al., 2015)



    – MPI-GE: Max Planck Institute for Meteorology Grand Ensemble (RCP8.5) (Maher et al., 2019)

Each individual Earth system model attempts to represent the actual Earth system. By using many members from multiple Large Ensembles, we can test the capabilities of the reconstruction methods across different model structures and states of internal variability. In other words, we test the reconstructions across a large set of plausible ocean states. Each ensemble member uses the same external forcing of historical atmospheric $CO_2$ before 2005 and Representative Concentration Pathway 8.5 (RCP8.5) afterwards. Internal variation in the ensemble members is caused by perturbing the initial state of the Earth

system at the point where that ensemble member is started. This causes each ensemble member to follow a unique trajectory of the ocean-atmosphere state over time. Each Testbed member consists of monthly averaged sea surface temperature (SST), sea surface salinity (SSS), Chlorophyl-a (Chl-a), mixed layer depth (MLD), $CO_2$ forcing (x$CO_2$), and p$CO_2$ each with a spatial resolution of $1°$x$1°$. Furthermore, each member of the LET assumes perfect coverage over space and time. That is, each p$CO_2$ estimate for a location is the true average for that location over that time period. This is compared to SOCAT data, which is

monthly, meaning that p$CO_2$ at a location could be from a single observation from that month. See Gloege et al. (2020) for additional details on the LET.

## 2.2   Data preparation

From each LET member, we extract monthly output for each of the six features listed above to be used in the ML models. Our analysis domain is restricted to the open ocean, defined here as having a depth greater than 100m.

In order to mimic real world observations, p$CO_2$ for each testbed member is sampled in space and time to match the SOCATv5 data product (Bakker et al., 2016; Sabine et al., 2013). As would be the case for the real world where actual SOCAT data would be the only p$CO_2$ data, the task for the ML approaches is to learn a representation of the system based on these limited observations and to then apply this representation to full-field driver data in order to upscale to the global ocean. The key advantage of the LET is that the true values for p$CO_2$ are known, and thus we can compare the reconstruction result to this

"unseen" data.

    The SOCAT sampled data is randomly split into 3 parts for each LET member: train (60%), validate (20%), and test (20%). Each member has a different random split so that we can understand how the approaches generalize from a wide array of training data points. The training set is used to fit each ML algorithm, the validation set is used to select tunable parameters and avoid over-fitting, and the test set is used for performance evaluation. Using the LET allows the non-SOCAT sampled

data to be leveraged for a final evaluation as well. We consider the unseen data to determine reconstruction ability, while the SOCAT-sampled test group allows us to understand what we would have seen with only real-world data.

    To eliminate outliers that could unduly influence the reconstruction, we eliminate all p$CO_2$ values above $816\mu$atm, the absolute maximum value in SOCATv5. We exclude these points as they are not representative of the real ocean; this exclusion primarily impacts the CanEMS2 model ($\sim$11.3K data points per member, or <0.1% of the data).





## 2.3 Machine learning approaches

We use three machine learning approaches to reconstruct surface ocean $pCO_2$ (the target variable) using features derived from SST, SSS, Chl-a, MLD, $xCO_2$, latitude, longitude, and time. The key characteristics and our development process are described here, with further details provided in Appendix A. For all approaches, we train the algorithms using mean squared error (L2 loss). The algorithms are implemented using Python library packages, as described below.

### 2.3.1 Feed-forward neural network (NN)

The NN is implemented using Keras with tensorflow 2.0 backend (Chollet et al., 2015). The main parameters considered for the NN are number of hidden layers and number of units per layer, which determine the size of the network. Additional parameters considered are learning rate, activation function, batch size, epochs, loss function regularization, and dropout. These parameters must be set by the user; they cannot be learned from the data. Learning rate controls the size of parameter updates at each step, and can be optimized during training. The activation function determines the non-linear transformation applied in the training process. Batch size represents how many data points are used at a time to update parameters and is bounded by available memory. The number of epochs determines how long the algorithm is trained; running for too many epochs can lead to overfitting. Loss function regularization and dropout both attempt to prevent overfitting. Regularization is used to force the NN to have smaller parameter values, and is commonly used in many ML algorithms. Dropout randomly deactivates part of the network during each training update to force the NN to learn multiple pathways for predicting the output.

Initially, we tested NN parameters on several members from each model and evaluated performance on the validation set to preserve the test set. Parameters such as learning rate and activation did not affect performance, so we used standard values for these. For learning rate, we used the Adam optimizer with 0.01 rate; for activation, we used the rectified linear unit (ReLU) function. In terms of regularization, dropout did not improve performance, but small L2 parameter regularization was used. Regularization led to a better fit and helped produce more consistent results. For the size of the network, once it was large enough (approximately two layers with 500 units in each), there was no noticeable difference in performance. We did not use a network with many layers, as used in deep learning, for this task. Deep learning is more applicable when there is enormous amounts of raw data and the depth of layers is required to extract intermediate representations of the data. In this case, we have more limited data with a clear set of feature variables.

For the NN, large variation in output occurred in different runs operating with the same architecture. The only difference in these cases was the random initialization. Because of this instability, we chose a single architecture and ran it five times for each LET ensemble member. From these five, the top two in terms of root mean squared error (RMSE) are evaluated and of these two, the one with the lower bias is identified as the best run. This best run is selected as the final NN output for that ensemble member.





### 2.3.2    Extreme gradient boosting (XGB)


The XGB approach (Chen and Guestrin, 2016) is implemented using the Scikit-Learn wrapper for XGBoost. The primary hyperparameters tuned were the maximum depth of each tree and the number of estimators, or trees, in the series. Maximum depth controls the complexity of each individual tree, while the choice of number of estimators balances a good fit without overfitting. We used cross validation over a grid of values to select these parameters. In preliminary tests on several members

from each model, the same optimal parameters were selected during the grid search. Because of this, instead of evaluating this grid for each member, we selected one member for each model and performed the grid search on this member. The optimal parameters were then applied to the other members for the model. Ultimately, the same parameters were selected in the grid search for all four models.

### 2.3.3    Random forest (RF)

The RF (Breiman, 2001) is implemented using the Scikit-Learn API. This approach followed a similar method as XGB. We tuned maximum depth and number of estimators hyperparameters using the same grid search method. One model selected different hyperparameter settings in the grid search than the other three.

### 2.4    Evaluation of reconstructions

We evaluate the ML approaches in three parts. First, we look at set of goodness-of-fit metrics to understand performance on

both the test and "unseen" data. Second, we use the full field input data to create complete $pCO_2$ reconstructions in space and time, and with this we can fianlly evaluate each approach's capability with respect to estimating all points.

After developing a general understanding of the fidelity of the ML approaches, we consider the degree to which the spatial and temporal patterns of $pCO_2$ can be reconstructed, following the approach of Gloege et al. (2020). We consider the spatial pattern of long-term mean bias. We decompose the $pCO_2$ from each ensemble member and the corresponding reconstruction

at every point for multiple time scales to assess the fidelity of the the phasing and amplitude of the reconstruction.

The temporal decomposition consists of breaking down the time series at each grid point into seasonal, sub-decadal, and decadal components. We perform this decomposition on the original modeled $pCO_2$ for each ensemble member and compare it to the ML model reconstruction for the same member using standard statistical metrics (Section 2.5).

Temporal decomposition is performed as follows. A linear-trend at each $1°x1°$ is removed at each grid cell to account for

the pseudo-linear increase in atmospheric $CO_2$ over 1982-2016. Second, a repeating annual seasonal cycle is calculated from the detrended time series. After subtracting the detrended and de-seasoned compoent at each location, the decadal signal is determined by applying a locally weighted regression (loess) smoother with a 10-year window. Finally, the residual signal is termed the sub-decadal component.



## 2.5 Statistical metrics

Reconstruction skill for both test and unseen data are evaluated on a series of summary metrics. These include root mean square error (RMSE), bias, mean absolute error, and maximum error.

**RMSE** is a measure of how well the predictions fit the data, with low values indicating a better fit. This penalizes larger errors much more severely in proportion to the squared error. For data points $y_i$ and predictions $\hat{y}_i$ corresponding to $i = 1, \ldots, N$, this is calculated as $\sqrt{\frac{1}{N} \sum_{i=1}^{N} (y_i - \hat{y}_i)^2}$.

**Bias** is calculated as the average error of the reconstruction $\frac{1}{N} \sum_{i=1}^{N} (y_i - \hat{y}_i)$. Bias is a measure of the systematic discrepancy between the reconstruction and model over the long term. It is important to note that values near zero may be misleading as positive and negative discrepancies can cancel each other out.

**Mean absolute error** (MAE) is $\frac{1}{N} \sum_{i=1}^{N} |y_i - \hat{y}_i|$. This metric provides an estimate of the average magnitude of the error, regardless of direction. While it does not show the systematic discrepancy, it shows how far off a prediction will be on average.

**Max error** identifies the largest absolute error value among the predictions, $\max_{i=1}^{N} |y_i - \hat{y}_i|$. While the previous metrics focused on average performance, this gives a sense of worst case performance.

We evaluate these for each individual ensemble member and then average across all ensemble members. The variance across the 100 members is also presented to determine the degree of consistency across the 100 members. To consider the spatial patterns of reconstruction skill, the same metrics are calculated at each geographical grid point, and then averaged across 225 members to understand the spatial performance.

For the temporal deconstruction, we focus on correlation and percent error of the standard deviation. These metrics are used to assess the degree to which the different ML approaches are able to reconstruct the temporal phasing and amplitude of variability in the original ensemble.

**Pearson correlation coefficient**, $r$ is the covariance between the reconstruction and the model divided by the product of 230 their standard deviations, $r = \frac{cov(R,M)}{\sigma_R \sigma_M}$. Correlation quantifies the synchrony between the reconstruction and model truth on various temporal scales. Values are bounded between $-1 \leq r \leq 1$, which quantifies the degree to which reconstruction captures the phasing observed in the model. Values near 1 and -1 indicate that the reconstruction and model are perfectly in or out of phase, respectively. Intermediate values indicate a phase shift between the two signals, with values closer to zero indicating a larger phase shift.

**Percent error** (% error $= \frac{\sigma_R - \sigma_M}{\sigma_M} * 100$) in the standard deviation quantifies the degree to which the reconstruction correctly captures the amplitude of $CO_2$ flux variability as observed in the ensemble member. This metric indicates whether the reconstruction overestimates (% error $> 0$), underestimates (% error $< 0$), or perfectly captures (% error $= 0$) the amplitude of variation in the original ensemble member.





**Table 1.** RMSE, standard deviation of RMSE ($\sigma_{RMSE}$), bias, MAE, max error, correlation, and amplitude % error for test and unseen data ($\mu$atm) averaged across members of LET. Bold represents the top performer(s) for that metric.

|  | Test data | | | | | | | Unseen data | | | | | | |
|---|---|---|---|---|---|---|---|---|---|---|---|---|---|---|
|  | RMSE | $\sigma_R$ | Bias | MAE | Max error | r | % error | RMSE | $\sigma_R$ | Bias | MAE | Max error | r | % error |
| NN | 8.47 | 1.03 | 0.10 | 5.82 | 158 | 0.98 | 2.48 | 14.64 | 2.48 | **0.00** | 9.07 | **481** | **0.93** | 5.02 |
| XGB | **5.66** | **0.91** | **0.00** | 3.69 | **132** | **0.99** | **1.13** | **14.07** | **1.95** | 0.13 | **8.57** | 487 | **0.93** | **4.19** |
| RF | 5.79 | 1.10 | 0.03 | **3.13** | 175 | 0.94 | 2.87 | 16.48 | 2.16 | 0.91 | 10.31 | 497 | 0.91 | 10.69 |

## 3 Results

### 3.1 Overall reconstruction performance

Table 1 shows performance on a suite of statistical metrics across all 100 members for each ML approach on both the test and "unseen" data. Figure 1 provides a visualization of the distribution across the 100 members for RMSE and the amount of degradation in performance from test to unseen data. All members have low average bias on the test set (NN: 0.10, XGB: 0.00, RF: 0.03$\mu$atm) and on the unseen set (NN: 0.00, XGB: 0.13, RF: 0.88$\mu$atm). On MAE, RF edges out XGB slightly on the test set (NN: 5.82, XGB: 3.69, RF: 3.13$\mu$atm), but XGB and NN both perform better on the unseen set (NN: 9.07, XGB: 8.57, RF: 10.31$\mu$atm) (Table 1).

The XGB produced the lowest error on the test data, with an average RMSE of 5.66$\mu atm$, but its performance suffered a large degradation, 148%, on extrapolation to the unseen data (RMSE = 14.07 $\mu$atm) (Figure 1). The RF performed similarly on the test data (RMSE =5.79 $\mu$atm); however, the RMSE degraded by 183% on the unseen data (RMSE = 16.48 $\mu$atm). Though NN had the highest test RMSE (8.47$\mu$atm) its performance on extrapolation to unseen data is comparable to the XGB (RMSE = 14.57 $\mu$atm), only degrading by 73%. In summary, XGB most closely fits the test data, but degrades substantially when extrapolated to unseen points. In contrast, while NN is less capable of fitting the test data, its performance degradation is much lower when extrapolated to unseen data (Figure 1).

While the XGB has the lowest average maximum error across members on the test data, all approaches have a similar maximum error on the unseen data. For this metric, degradation from test to unseen data in the NN model is 203%, 266% in the XGB model, and 183% in the RF model. RF has the least relative degradation because it has the worst performance on the test data to begin with.

### 3.2 Mean spatial performance

Though the global mean bias is near zero for all three approaches (Table 1), there are regions of positive and negative bias across the globe in each (Figure 2). The patterns of bias are quite similar across the methods, but the severity differs. XGB has fairly low average bias in the open ocean, RF has the largest biases, and NN has moderate biases. Open ocean biases are overall lower in the Northern Hemisphere.





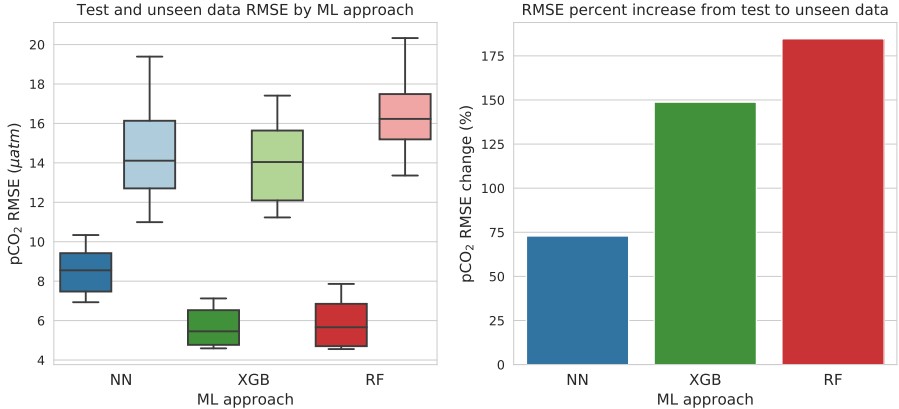

**Figure 1.** Left: boxplot of RMSE for 100 members of the LET. RMSE between reconstruction and original LET ensemble member by ML approach for (dark) test data and (light) unseen data: (blue) NN, (green) XGB, (red) RF. Right: RMSE degradation from test to unseen data sets by ML approach: (blue) NN, (green) XGB, (red) RF.

Across the 100 members for each approach, the spread of the bias indicates how consistently the ML method is able to accurately reconstruct $pCO_2$ (Table 1). Figure 3 illustrates that by showing the variance of the bias at each point across the 100
members. The locations where mean bias is small (Figure 2) are also where variance is low. This indicates that the ability of the reconstruction to provide a low-bias result is robust across the ensemble members at these locations.

Regions of high bias are consistent across the ML methods (Figure 2). Coastal regions are generally poorly reconstructed, since the reconstructions are based on primarily open ocean data. The southeast Pacific is a region of notable bias and variance for all three methods, and the Southern Ocean has more bias and variance of bias (Figure 3) than the Northern extratropics.

**3.3    Temporal decomposition**

At each point, the reconstruction is divided into three temporal components: seasonal, sub-decadal, and decadal. For the deconstructions, we evaluate performance using correlation (Figure 4) and percent error of the amplitude (Figure 5).

On the seasonal timescale, all of the approaches capture the phasing well with $> 0.9$ correlation across most of the ocean (Figure 4, top row). The globally averaged correlation is $> 0.9$ for all methods (Table A1). Phasing performance drops off for
the decadal time scale, with some portions of the western Pacific and Northern Ocean above $0.8$ correlation and large swaths between $0.6 - 0.8$ (Figure 4, bottom row). NN is slightly better than XGB at capturing decadal phasing in large open ocean regions in both the Northern and Southern Hemisphere, and thus for the global average, correlation is higher for NN on the decadal timescale ($r_{NN} = 0.71$, $r_{XGB} = 0.64$; Table A1). For the sub-decadal timescale, phasing is only very well captured ($> 0.8$) in the western equatorial Pacific, and in the subtropics there are swaths of moderate phasing skill ($0.6 - 0.8$) (Figure 4,
middle row).





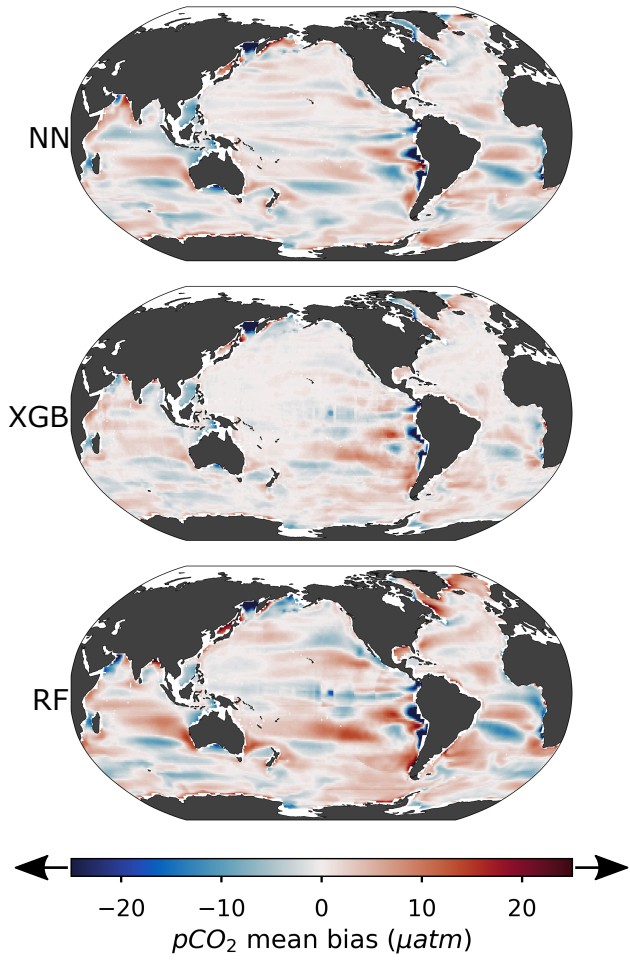

**Figure 2.** Average bias between reconstruction and original LET ensemble member by ML approach: (top) NN, (middle) XGB, (bottom) RF. Arrows indicate that values beyond limits are capped for visualization. Table A1 provides global and regional average results.

For seasonal amplitude (Figure 5, top row), NN and XGB generally provide high fidelity reconstructions, but RF systematically underestimates seasonal amplitudes. All approaches have a tendency to overestimate the decadal amplitude except in the western equatorial Pacific (Figure 5, bottom row). Figure 6 provides a comparison of the reconstructed global mean decadal amplitude to the actual amplitude. NN is closest to the true amplitude overall, and does particularly well when amplitudes are smaller. NN tends to underestimate decadal amplitudes when actual amplitudes are large. RF significantly overestimates the amplitude of decadal variability in almost all ensemble members. For members with smaller decadal variability, XGB tends


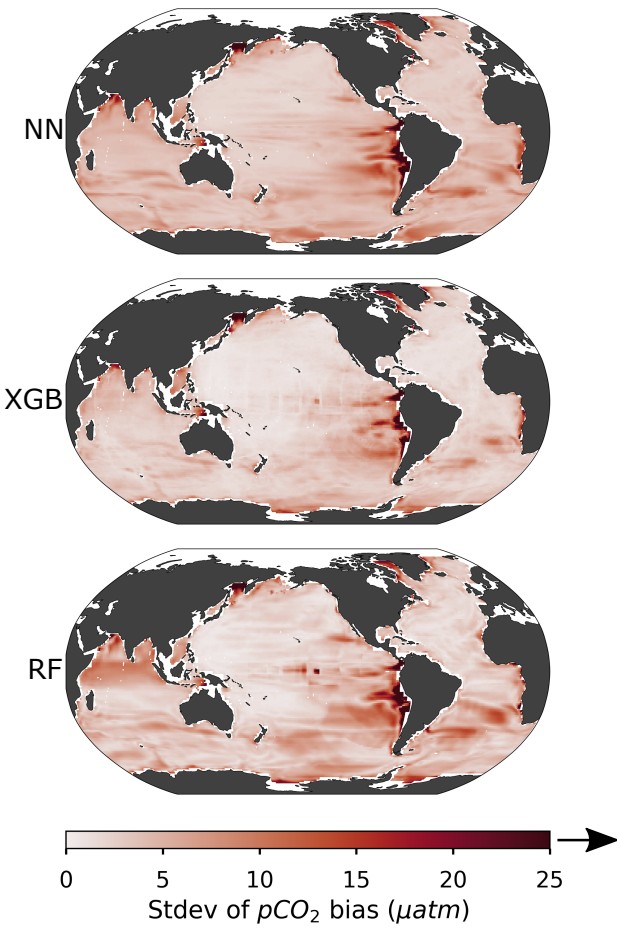

**Figure 3.** Standard deviation of bias across the LET members by ML approach: (top) NN, (middle) XGB, (bottom) RF. Arrows indicate that values beyond limits are capped for visualization.

to overestimate amplitude, but underestimates for members with large decadal amplitude. Globally averaged, NN has a 13% overestimate, while XBG overestimates by 28%, and RF by 60% (Table A1).

For the sub-decadal timescale, NN underestimates amplitude almost everywhere (global average -10%, Table A1). XGB and RF tend to overestimate sub-decadal amplitude in the subtropics, but underestimate it elsewhere (Figure 5, middle row).

### 3.4 Sampling vs. Mean spatial performances

As shown above, there are certain areas where all three ML approaches struggle to varying degrees. Considering bias versus the availability of test data (Figure 7), it can be seen how data availability drives prediction performance. Each point represents

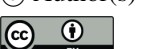

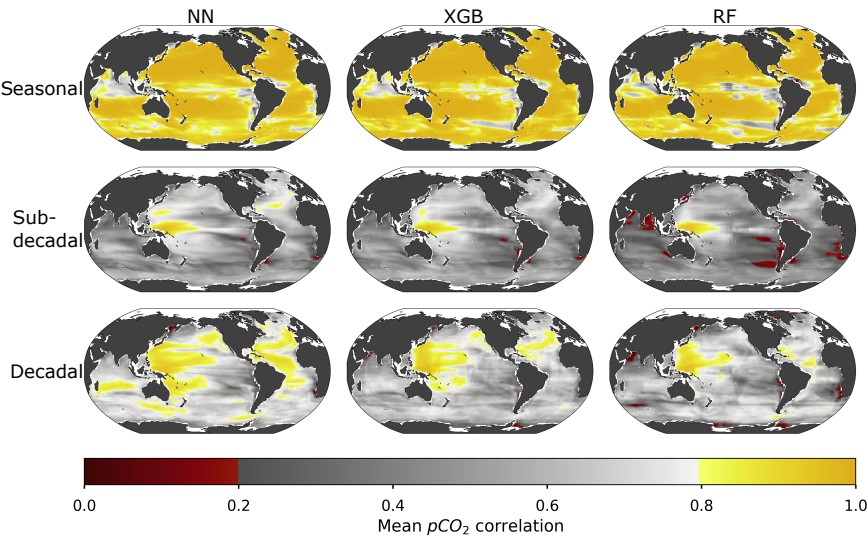

**Figure 4.** Average correlation at seasonal, sub-decadal, and decadal timescales across the LET by ML approach: (left) NN, (middle) XGB, (right) RF; (top) seasonal, (middle) sub-decadal, (bottom) decadal. Table A1 provides global and regional average results.

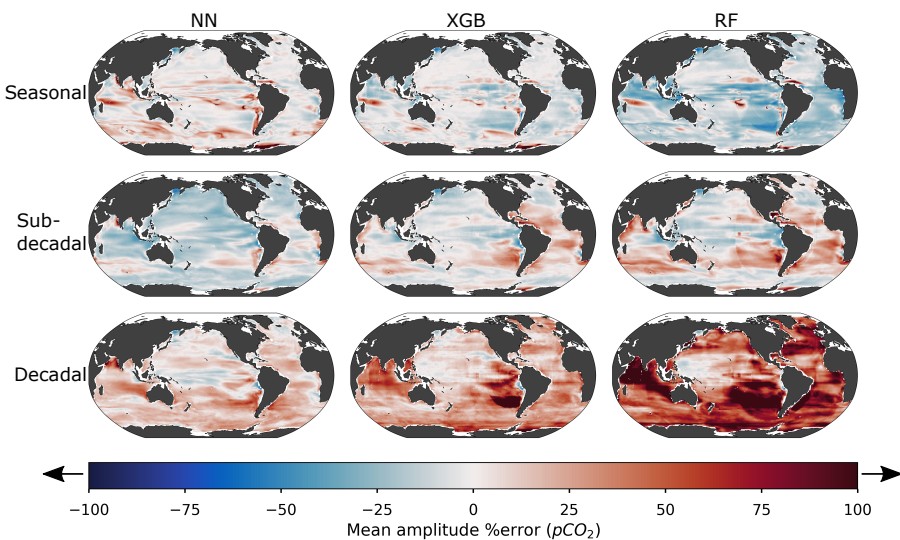

**Figure 5.** Mean % error of the amplitude at seasonal, sub-decadal, and decadal time scales per location across the LET by ML approach: (left) NN, (middle) XGB, (right) RF; (top) seasonal, (middle) sub-decadal, (bottom) decadal. Arrows indicate that values beyond limits are capped for visualization. Table A1 provides global and regional average results.

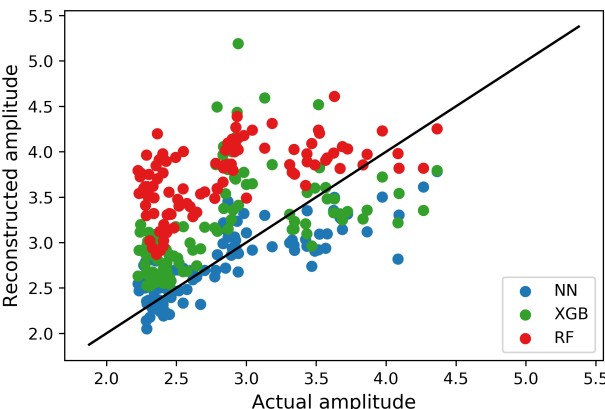

**Figure 6.** Reconstructed global average decadal amplitude vs. actual amplitude for each member across the ML approaches: (blue) NN, (green) XGB, (red) RF. Points above the black line overestimate the decadal amplitude, while those below underestimate it.

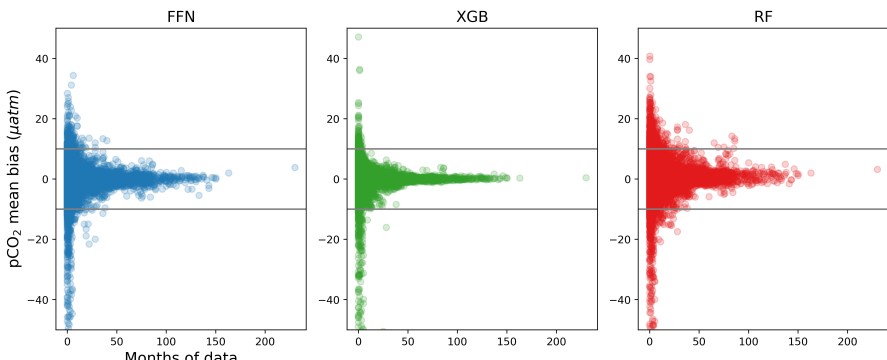

**Figure 7.** Average bias per location across the LET versus number of months of data available at test data locations. Each data point represents the mean bias at a geographical coordinate across the 100 members. Thin lines are at 10 $\mu$atm to facilitate comparison. Points with bias less than $-50\mu$atm (<0.3% of data) are excluded to improve presentation.

the time-average bias at a given location, averaged across the 100 members of the LET. When only sparse data are available for

a given location, predictions can be poor in all methods. However, as more months of data become available, there is a sharp decrease in mean bias. XGB reduces bias more quickly than the other two approaches. In other words, XGB performs best when only moderate amounts of data are available, as shown also with Table 1 and Figure 1.

## 4   Discussion

Overall, XGB best reconstructs the pCO$_2$ field. It has low bias on the test set, and the lowest RMSE on both the test and unseen

data sets. It is also most consistent across the 100 members in terms of RMSE, i.e. it has a low standard deviation of RMSE





across the members (Table 1) and a low standard deviation of bias (Figure 3). NN provides comparably average performance; however, this comes with higher variance in its performance (Table 1, Figure 3). Though RF does almost as well as XGB on the test data, it generalizes poorly. Gregor et al. (2019) also found that the random forest method performs poorly in pCO$_2$ reconstruction.

It is important to note that the NN model does not consistently perform well, and steps must be taken in implementation to limit the spread of its performance. For this implementation, for each of the 100 ensemble members, we trained this algorithm five times and selected the best run, as in Denvil-Sommer et al. (2019). This reduces the resulting variance, but the fidelity of the NN result still has substantial spread (Table 1, Figure 1). The inherent instability of the NN approach is thus a concern for any real-world application. We find variation across NN runs for all sizes of networks considered. Running the model as 310 many times as possible can help to address this problem, but this iterative approach does add substantially to the already high computational cost of developing a NN algorithm.

Between the tree-based models, the superior performance of XGB on the unseen data compared to RF is attributable to the XGB algorithm focusing on previous errors when constructing the series of trees. RF trees are correlated, meaning that there is less ability to drive down the error of the fit. The greater fidelity of XGB here is consistent with its rapid adoption for many 315 commercial applications in recent years (Chen and Guestrin, 2016).

A significant benefit of the LET is the ability to simulate the model training process with real-world sampled data and then to directly examine performance on unseen data. The degradation in the various statistical metrics can provide useful quantification of the suspected degradation that occurs with these ML methods when they are applied to real-world data. Depending on the approach, the RMSE from test to unseen data can increase by 75%-185% (Figure 1). Recent related research 320 (Landschützer et al., 2013; Gregor et al., 2019) studying the degradation between SOCAT test data and independent pCO$_2$ data from the Lamont-Doherty Earth Observatory (LDEO) database (Takahashi et al., 2009) found similar levels of RMSE degradation with SOM-FFN (115%) and the ensemble CSIR-ML6 (56%). These results indicate that the comparison to the independent LDEO data is an accurate indicator of the aggregate level of degradation with extrapolation.

In our tests, the NN suffered the least degradation in performance from test to unseen data, indicating that it captures the 325 pCO$_2$ effectively without overfitting too much on the training data. Still, it is important to note that the greater degradation in XGB as opposed to NN does not mean NN performs better overall. For our implementation, XGB performance is high enough on the test data that even a larger drop off puts it in a better position on the unseen data than its competitors. Gregor et al. (2019) also found that the NN had the smallest difference in RMSE performance when compared to indpenedent data, while the ERT (similar to our RF) suffered the most degradation.

330 SOCAT pCO$_2$ data are collected along ship tracks and have substantial spatial along-track correlation (Jones et al., 2012), so the data are not entirely random and thus their value to ML method training is compromised. Gregor et al. (2019) ran an experiment comparing random sampling vs. sampling by year to separate training and test data. This approach reduces correlations between the training and test data because it segregates entire ship tracks to either the training or the test set. When data are randomly sorted, then training and test data will be from the same ship tracks and thus have greater correlation. Gregor





et al. (2019) find increased RMSE against the test set for this approach, and conclude that this is a more robust estimate of the true error. They recommend this sampling method in order to prevent overfitting.

As discussed by Gregor et al. (2019) and Gloege et al. (2020), there are many regions of the global ocean pCO$_2$ state space that are inadequately sampled. Gregor et al. (2019) propose that the convergence across recent implementation of ML methods and other approaches to pCO$_2$ extrapolation indicates that the available real-world sampling is essentially a "wall" that prevents further improvement of pCO$_2$ reconstruction skill. That is, reconstruction approaches may not be able to improve significantly from the current best-in-class because of prohibitive data sparsity.

To test the value of additional data using the LET, Gloege et al. (2020) add a modest amount of additional sampling in the Southern Ocean to SOM-FFN. They show that this could substantially improve the ability of this neural network to reconstruct decadal timescale variability, both in the Southern Ocean and also across the global ocean. As seen here in Figure 7, the number of months of data at a given location is a significant driver of performance for our ML methods. XGB performance with moderately sparse data is the most impressive. As soon as the XGB model has a small amount of data at a location, the range of bias is much reduced compared to the other ML methods. This indicates that additional sampling in low coverage regions combined with optimized ML algorithms could lead to improved pCO$_2$ reconstructions going forward. It would be valuable to repeat the test of Gloege et al. (2020) with the XGB model to determine how much a modest increase in Southern Ocean sampling could improve Southern Ocean and global performance.

While on average the XGB and NN models reconstruct pCO$_2$ with low bias and RMSE in the open ocean, there are clearly regions where these methods and the RF do not work well. Of note are eastern boundary upwelling regions and the Southern Ocean. The very limited sampling for these regions is consistent with the state space for pCO$_2$ being inadequately represented in the data available for algorithm training. For similar reasons, algorithms built specifically for the coastal regions also appear to be needed (Roobaert et al., 2019; Laruelle et al., 2017). Specialized approaches could focus on localized patterns or incorporating additional information into the training process to achieve a better representation of these areas.

Another consideration is the timescale of interest. We find very good performance seasonally, but deteriorating performance on the sub-decadal and decadal timescales. Seasonal variations are large in most locations, and are also the best-sampled given our dataset from 1982-2016. Thus, these are the easiest signals for all three ML approaches to identify. However, the NN shows stronger performance in the decadal time series — the one place where it outshines XGB.

Why does the NN perform best for the phasing and amplitude of pCO$_2$ variability on decadal timescales? In Figure 1, we show that despite not fitting the test data as well, NN performs best in generalization to unseen data. XGB and RF have more trouble extrapolating. Relative to seasonal variation, the decadal timescale is very poorly sampled, with the entire dataset extending only 35 years. It is likely that the fundamental structures of these models causes this difference. The NN creates a non-linear mapping of input to output, without creating distinct regions in the feature space. Conversely, the tree-based models do create distinct regions, effectively chopping up the training data into distinct bins. Given the limited sampling of the geophysical state space relevant to pCO$_2$ variations on decadal timescales, the tree based methods perform more poorly in this extrapolation. The ability of NN to generalize beyond the training set, as indicated by Figure 1, appears to be the reason for its ability to better construct decadal timescale variations.





Though NN generalizes better, we find that XGB performs so well against training data that its overall performance is slightly better than NN (Table 1 and Figure 1). This is likely due to how this algorithm constructs its estimators. The initial trees in the series are very weak, but they still capture the rough $pCO_2$ trends across the globe. As more trees are added, the algorithm focuses on reducing errors, particularly in locations that are well sampled. This leads to an excellent fit in areas with sufficient sampling while still capturing the average trends elsewhere. It is also worth noting that XGB is more computationally

efficient than the NN, a considerable benefit for use with datasets of this size.

    Gregor et al. (2019) propose the use of an ensemble of ML methods for reconstruction of surface ocean $pCO_2$, and included NN and XGB methods in their final ensemble. Our results suggest that in their CSIR-ML6 ensemble, XGB helps to strengthen overall performance while the NN enhances the ability to generalize to features and timescales that are poorly represented in the training data.

**5   Conclusions**

We apply the Large Ensemble Testbed (LET) to test the performance of three machine learning (ML) approaches in extrapolating sparse ($\sim 2\%$ coverage) surface ocean $pCO_2$ observations to global coverage. Using 25 members from each of four Earth System Models, we assess the performance of three common ML approaches: Feed forward neural network (NN), eXtreme Gradient Boost (XGB), and Random Forest (RF).

Analyzing the performance of the models on test and unseen data, we find that XGB produces the best $pCO_2$ reconstruction overall, with low variance across all ensemble members and strong performance on the unseen dataset. The NN can also provide a high-quality reconstruction, but its performance is highly variable. The high variance is partially mitigated with an iterative training and selection approach. Certain geographic areas are poorly reconstructed by all three approaches, particularly coastal areas, eastern boundary upwelling zones and the Southern Ocean. For timescales, we find seasonality to be well-

reconstructed by all approaches, but that sub-decadal and decadal timescales can be robustly captured in only a few regions. Decadal variability is of particular interest to the ocean carbon cycle community (Landschützer et al., 2015; Gruber et al., 2019), and we find that NN performs slightly better for the decadal timescale than does XGB (Figure 4). The creation of a non-linear mapping, without creating distinct regions in the feature space, appears to lead the NN to better extrapolate to the poorly-sampled decadal timescale.

Data sparsity remains the dominant challenge for accurate reconstruction of the global $pCO_2$ field, and thus the global ocean $CO_2$ flux, from observations. Using the LET, Gloege et al. (2020) show that adding a small number of additional samples in the Southern Ocean would substantially improve SOM-FFN performance. Our findings indicate that targeted additional sampling to fill in the state space that relates $pCO_2$ to its driver variables and XGB algorithms would be a promising path forward to higher fidelity $pCO_2$ reconstructions.

The LET provides an excellent testing ground for future work on $pCO_2$ reconstructions. Developing a deeper understanding of the regions of $pCO_2$ and driver variable state space that are not yet adequately sampled could help to target new observation programs. Future work could explore additional ML methods or tunings, and a more exhaustive inquiry into the variability of





results across the 100 ensembles could improve knowledge of method strengths and weaknesses. Comparison of these results
to the same approaches applied after clustering the ocean into basin-scale regions, as implemented previously (Landschützer
et al., 2013, 2014; Gregor et al., 2019), would improve understanding of the impacts of clustering on performance.

More broadly, when exploring ML approaches for spatial extrapolation of geophysical datasets, tests using synthetic data
from an Earth system model can provide valuable insight. Independent data sets are typically too sparse and too precious to set
aside for a comprehensive evaluation. For our application to surface ocean $pCO_2$, findings with respect to skill are consistent
with previous assessments using independent data (Landschützer et al., 2013; Gregor et al., 2019). The LET compliments these
assessments by providing a more holistic evaluation of performance across timescales and for the full global domain.

*Code and data availability.* The 100 member large ensemble testbed is publicly available at https://figshare.com/collections/Large_ensemble_
pCO2_testbed/4568555, and is managed by L.G. $pCO_2$ reconstructions are publicly available at https://doi.org/10.6084/m9.figshare.c.5122601.
v1. Data analysis scripts are contained in GitHub repository https://github.com/jstamell/ML_for_ocean_pCO2_interpolation. SOCATv5 is
available at https://www.socat.info/index.php/previous-versions/. ERA-interm 6 hourly output is available at https://apps.ecmwf.int/datasets/.
Any other inquiries should be addressed to J.S.

## Appendix A: Methods - additional detail

### A1 Data processing

Additional detail on the features used is available here:

- **Latitude/longitude:** Primary transformation applies a sine function to latitude. Additional transformations combine
sine/cosine transformations for both coordinates. (Denvil-Sommer et al., 2019)

- **Date:** Sine and cosine functions are applied with an annual period to create repeating cycles for time.

- **Sea surface salinity (SSS):** Calculate an additional anomaly feature which subtracts the annual mean from each point.

- **Sea surface temperature (SST):** De-trend SST over time by calculating a linear trend at each data point. Calculate an
  additional anomaly feature which subtracts the annual mean from each point.

- **Chlorophyll-a (Chl):** Apply a log transformation. Calculate an additional anomaly feature which subtracts the annual
  mean from each point.

- **Mixed layer depth (MLD):** Calculate a climatology at each point, which looks at the average monthly value for each
  lat/lon point over the dataset. Apply a log transformation to this climatology.

- **$CO_2$ forcing:** No transformation

- **$pCO_2$:** No transformation





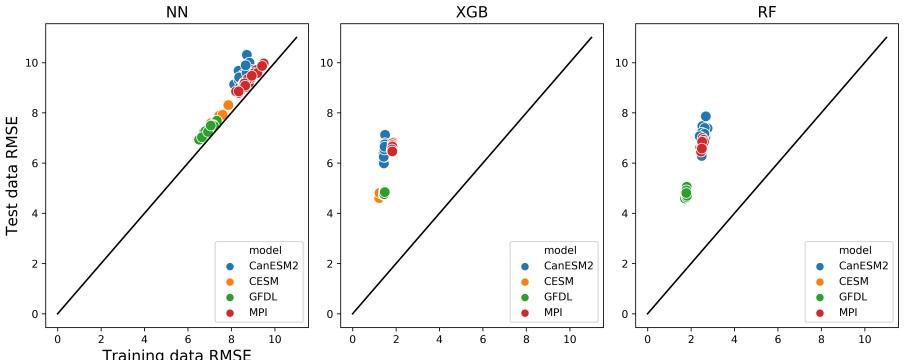

**Figure A1.** Training vs. test set RMSE for all members in the LET for each ML approach. Colors represent which ensemble model the member is from.

## A2 ML model training

For each ensemble member, model points corresponding to SOCATv5 (Bakker et al., 2016) open ocean sampling are randomly split into three parts: 60% training, 20% validation, 20% test. This allows us to simulate how we would train a ML model with real world data. All other points, or "unseen" data, are used to understand how the ML method extrapolates. We expect
deterioration in performance as compared to the SOCATv5-sampled data set due to correlations in the data along ship tracks (Jones et al., 2012). Previous work has provided ample evidence that the full state space for pCO$_2$ in the open ocean across the last several decades is not sufficiently sampled (Gregor et al., 2019; Gloege et al., 2020).

An alternative to random sampling of the dataset would be to randomly sample ship tracts or years. This could better adjust for the correlation in the data and close the error estimation gap from test to unseen data. Future work with the LET should
explore this approach.

Figure A1 compares the training and test RMSE data for each approach. The training RMSE for the NN is very similar to the test data, indicating that the NN is learning the patterns in the data efficiently without overfitting the training data. The tree based models achieve a lower training RMSE than test RMSE. This is unsurprising because they are constructed to learn the average values of the training data in the regions of the state space that they create. In fact, XGB is designed to eventually
achieve a training RMSE of 0 if run for long enough. Though training data RMSE is lower than test RMSE, this does not mean we are overfitting and sacrificing test set performance. Instead, test set performance is actually lower for both XGB and RF, as also shown in Table 1 and Figure 1. Our use of cross validation against a validation set when the number of estimators are selected helps to guard against overfitting.

When training the ML models, we use L2 loss (mean squared error) to fit the models. In the tree-based models, the validation
set is used for one member from each of the four Earth System Models in the LET to tune hyper-parameters (e.g. number of estimators). Then, all members for that LET model are trained using these hyper-parameters. The idea behind this is that





optimal setting for these models could vary between LET models, but are not likely to vary as much within the models. In general, the hyperparameter settings for RF and XGB did not affect performance greatly.

For NN, the architecture (e.g. number of hidden layers, number of hidden units) did not drive a large variation in performance. However, for a given architecture, different random initialization does cause high variance in performance. Regularization helps to reduce this variance somewhat. To further reduce variance, for each LET member, we run 5 neural networks with the same settings and use the validation set to select the best one, applying criteria as described in the main text. Applying dropout between the layers did not reduce the variation.

We use the Keras/tensorflow backend python implementation for the NN. For this model, we use a 2 hidden layer network with 500 units each and the RELU activation function. We use the Adam optimizer with a 0.01 learning rate and light L2 regularization.

We use the XGBoost python implementation for the XGB model. We use default settings except for number of estimators and max tree depth. For these, we consider a grid of [2000, 3000, 4000] estimators by [4, 5, 6] depth. From this grid, the 4000 estimator and 6 depth model was selected in cross-validation for all four Large Ensembles.

We use the scikit-learn python implementation for the RF model. We use default settings except for number of estimators and max tree depth. For these, we consider a grid of [50, 100, 200] estimators by [30, 40] depth From this grid, the 200 estimator and 40 depth model was selected in cross-validation for all three of the Earth System Models. For CanESM2, we use 30 depth.

### A3 Regionally and globally averaged metrics

In Table A1, we present regional and global averages of long-term bias (Figure A3) and for correlations (Figure 4) and percent amplitude error for each timescale (Figure 5).

### A4 Ensemble variation

While this paper focuses on the differences between ML approaches, there is also variation across the ensemble models. Figure A2 provides an overview of the RMSE for each approach, broken down by model, for the test and "unseen" data sets. For each model, the overall conclusions discussed in the paper are the same, the most significant finding being that across all three ML approaches, XGB has the highest overall performance. NN is also very competitive. Notably, however, the ML algorithms consistently yield larger errors in reconstructing the CanESM2 and MPI model fields, compared to the other two models. Indeed, there are underlying features of these two models that materialize high error in the reconstructions. The MPI model has larger variance of climate variables as compared to the others, and the CanESM2 model generally has high $pCO_2$ values in some locations, even after filtering out extreme values ($pCO_2$ values $> 816 \mu$atm). Interestingly, reconstruction performance, as reflected by RMSE, degraded least in the MPI model, followed by CanESM2 for all ML approaches.

Figure A3 shows the average bias for each approach, broken down by each ensemble model. As we would expect from Figure A2, the ML approaches perform better on the GFDL and MPI models. Looking across the approaches for each model reinforces that the XGB performs the best at capturing the $pCO_2$ field with low bias, particularly in the open ocean. While the magnitude of the bias is different across the ensembles, the locations where they struggle are similar.





**Table A1.** Mean bias for the full reconstruction, split into three latitude regions. Average correlation and % error of the amplitude for seasonal, sub-decadal, and decadal timescales, split into three latitude regions.

| Metric | Data | Approach | Latitude | | | Global |
| --- | --- | --- | --- | --- | --- | --- |
| | | | >35°N | 35°S - 35°N | <35°S | |
| Mean bias | Full reconstruction | NN | -0.67 | -0.42 | 0.44 | -0.13 |
| | | XG | -1.06 | 0.06 | 0.38 | 0.04 |
| | | RF | 0.59 | 0.53 | 1.10 | 0.75 |
| Correlation | Seasonal | NN | 0.96 | 0.91 | 0.89 | 0.91 |
| | | XG | 0.97 | 0.92 | 0.91 | 0.92 |
| | | RF | 0.95 | 0.90 | 0.88 | 0.90 |
| | Sub-decadal | NN | 0.64 | 0.58 | 0.50 | 0.56 |
| | | XG | 0.61 | 0.55 | 0.50 | 0.54 |
| | | RF | 0.52 | 0.45 | 0.44 | 0.46 |
| | Decadal | NN | 0.68 | 0.73 | 0.70 | 0.71 |
| | | XG | 0.66 | 0.67 | 0.60 | 0.64 |
| | | RF | 0.59 | 0.62 | 0.59 | 0.61 |
| % error of amplitude | Seasonal | NN | -2.3% | 2.4% | 12.1% | 5.4% |
| | | XG | -4.5% | -3.5% | 0.9% | -2.0% |
| | | RF | -12.7% | -13.4% | -12.9% | -13.1% |
| | Sub-decadal | NN | -17.9 | -10.9% | -7.0% | -10.3% |
| | | XG | -6.8% | 2.7% | 3.3% | 1.7% |
| | | RF | -3.1% | 6.6% | 7.3% | 5.6% |
| | Decadal | NN | 2.1% | 11.3% | 18.1% | 12.6% |
| | | XG | 17.2% | 26.6% | 34.1% | 28.2% |
| | | RF | 56.9% | 55.4% | 67.2% | 59.9% |

The differences across these models highlight some of the challenges for reconstruction methods. By nature, all ML algorithms will struggle to produce a successful reconstruction when there is high uncertainty in the data, which is clearly exhibited by the poor performance of each approach in areas such as coastal regions, segments of the Southern Ocean, and other areas of particular data sparsity and high variability. To accommodate for these outliers while attempting to retain data integrity, we utilized minimal data cleaning (i.e. only removing extreme outliers). One potential future exploration with the LET would be
to assess more fully the impact of data validation procedures on algorithm performance.





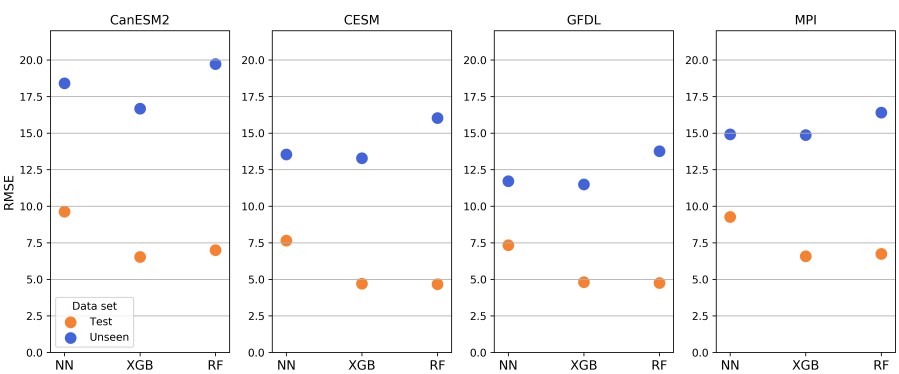

**Figure A2.** Test and unseen set RMSE averaged across members for each ensemble model in the LET for each ML approach.

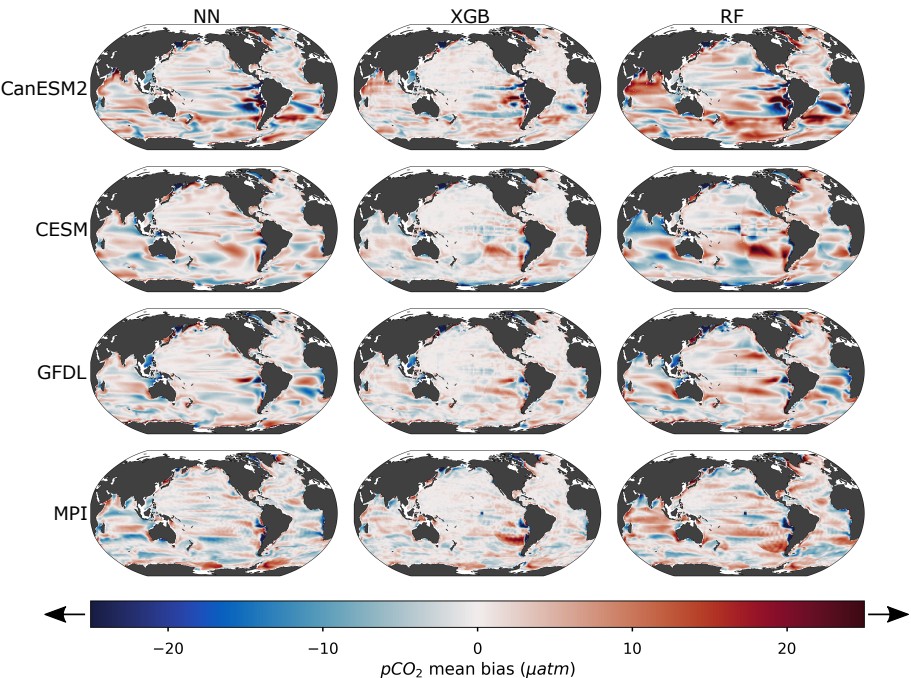

**Figure A3.** Average bias between reconstruction and original LET ensemble member by ML approach and ensemble model: (left) NN, (middle) XGB, (right) RF; (top) CanESM2, (second from top) CESM, (second from bottom) GFDL, (bottom) MPI. Arrows indicate that values beyond limits are capped for visualization.





*Author contributions.* L.G. and G.A.M. designed the study. J.S. prepared the data, trained the ML models, and created the reconstructions. R.R. assisted with the XGB training. J.S. and R.R. drafted the manuscript and performed all analyses. All authors (J.S., R.R., L.G., and G.A.M.) participated in the interpretation of the analysis, discussed results, and refined the manuscript.

*Competing interests.* The authors are not aware of any competing interest.

*Acknowledgements.* We acknowledge support for this research from Columbia University through the Department of Earth and Environmental Science and the Data Science Institute. The Surface Ocean $CO_2$ Atlas (SOCAT) is an international effort, endorsed by the International Ocean Carbon Coordination Project (IOCCP), the Surface Ocean Lower Atmosphere Study (SOLAS) and the Integrated Marine Biosphere Research (IMBeR) program, to deliver a uniformly quality-controlled surface ocean $CO_2$ database. The many researchers and funding agencies responsible for the collection of data and quality control are thanked for their contributions to SOCAT. We also thank the many scientists

who contributed to the development of the Earth System Model Large Ensembles used here.



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
