# Peer review of "Strengths and weaknesses of three Machine Learning methods for pCO2 interpolation"

_Geoscientific Model Development, 2020_

## Referee Comment (RC1) · Anonymous Referee #1 · 18 Nov 2020

Comments:
Stamell et al. compared three machine learning methods in interpolating surface pCO2 in the global ocean. The manuscript is lack of novelty and suffers many technical difficulties.

Major comments:
1) In essence, the authors developed three pCO2 models using three machine learning (ML) approaches (NN, RF, and XGB), and evaluated their overall performance in monitoring the seasonal, sub-decadal, decadal variabilities of surface pCO2. There are too many such studies in the published literature. In fact, the three machine learning (ML) methods presented in this study were already compared in Gregor et al. (2019), in which 6 supervised ML methods (including the three used in this study) were applied to reconstruct global surface pCO2. The authors concluded that all methods had overestimation in the reconstructed pCO2, yet Gloege et al (2020) also found overestimation using NN. So what is the overall novelty and significance of the present study?

2) There are lots of technical difficulties. The authors stated that Large Ensemble Testbed (LET) consists of 100 members across four initial-condition ensemble models, and each member is a representation of the real ocean climate system. In my understanding, data from these members are actually from the Earth system models. How accurate are these modeled data particularly those (SST, SSS, MLD, Chl-a, .etc) used to train the pCO2 model? How accurate are the pCO2 from the LET comparing to the *in situ* observations (SOCAT v5)? Without any evaluation, it is questionable to say these modeled data represent the real ocean system. I am not an expert on Earth system models, but why the authors say '100-member LET consists of 25 randomly selected member …' (L114-115)? What is difference between these members? The authors argued that the use of many members was to test the reconstruction capabilities of the ML across different ocean states, however, what is the impact of ocean state differences on the reconstructed pCO2? Also, there are some technical words that are quite difficult to follow without clear explanation (e.g., full field driver data, unseen data, LET). The ML was trained based on grid data at 1 by 1 degree, what is the impact of real spatial variability within the grid on the uncertainties of the reconstructed pCO2?

3) As to the overall structure of the manuscript, the authors presented details of the three ML methods in both Introduction and Methods. The earth system modeled data and SOCATv5 data are not well described, for example, the data coverage both spatially and temporally, and why they are used. In the ML approaches, again, why these three approaches were selected?

Specific comment:
L244: Statistics to the 'unseen' data is different from those listed in Table 1.

---

## Referee Comment (RC2) · Anonymous Referee #2 · 19 Jan 2021

This work by Stamell et al. compares the performance of three machine learning approaches, i.e., feed forward neural network (NN), XGBoost (XGB) and random forest (RF), based on the Large Ensemble Testbed. The authors did a lot of work, however, there are many unclear parts in the manuscript.

Major comments:

1. The literature review in this manuscript only mentioned previous studies using SOM-FFN to interpolate the $pCO_2$ field. What about the other methods, especially the three methods tested in this study? Have they been used in estimating $pCO_2$ field before? What are the major improvements of this study?
2. The SOM-FNN performed well in interpolating the $pCO_2$ field, but is likely to overestimate in the Southern Ocean. Is this issue improved in the three methods from this study?
3. I am a little confused about the data used to test the three ML methods. What are the target data or ground truth data when training the model? The data from the Large Ensemble Testbed (LET) are the ensemble of Earth system models, which are not observational data. While the SOCATv5 data product, which are actual measurements data, seems not to be included in the model training. Please clarify.
4. How are the train (60%), validate (20%) and test data (20%) split? Are they spatial-temporal randomly divided, or according to the locations or times? Different split methods lead to the evaluation of different model abilities. Split according to locations indicates the model's ability in spatial interpolation, while split according to times indicates the model's ability in temporal prediction. Please clarify.

Minor comments:

What is the sample size of the data?

Line 196: "fianlly" should be "finally"

---

## Author Comment (AC1) · 4 Feb 2021

Reviewer 1 Comments:

Stamell et al. compared three machine learning methods in interpolating surface pCO2 in the global ocean. The manuscript is lack of novelty and suffers many technical difficulties. Major comments:

*1) In essence, the authors developed three pCO2 models using three machine learning (ML) approaches (NN, RF, and XGB), and evaluated their overall performance in monitoring the seasonal, sub-decadal, decadal variabilities of surface pCO2. There are too many such studies in the published literature. In fact, the three machine learning (ML) methods presented in this study were already compared in Gregor et al. (2019), in*

*which 6 supervised ML methods (including the three used in this study) were applied to reconstruct global surface pCO2. The authors concluded that all methods had overestimation in the reconstructed pCO2, yet Gloege et al (2020) also found overestimation using NN. So what is the overall novelty and significance of the present study?*

Thank you for this comment.

The reviewer is correct that these types of approaches have been applied to pCO2 reconstruction in past studies such as Gregor et al. (2019). However, these past studies have not be able to quantify extrapolation uncertainty. Gregor et al. (2019) attempted to resolve this issue using independent data sets; however, these data sets still suffered from data sparsity issues. Because we work on simulated pCO2 from the Large Ensemble Testbed, we add the ability to assess uncertainties in extrapolation beyond the test data, or "unobserved data" in our updated terminology.

Yes, Gloege et al. (2020) also show overestimation of decadal variability with a specific NN implementation, the SOM-FFN (Landschutzer et al. 2016). For the NN in this paper, we make different choices, such as not breaking the ocean up into biomes. There are other NNs, such as in Gregor et al. (2019) and in CMEMS (Denvil-Sommier et al. 2019) that make their own independent choices. Our goal is to provide a reasonable basis for comparison between the NN, RF and XGB methodologies. Our implementations all use the same input driver data and do not use spatial clustering into biomes so that they are as simple as possible. Our focus is on understanding the extrapolation skill across different methodologies - something that has not been presented in the literature before. For example, Gregor et al. (2019) combine different methodologies to create an ensemble method, but do not explore in detail the strengths and weakness of the individual methods.

We have added text to the abstract and text to clarify that the goal of this work is to quantify extrapolation uncertainty for NN, RF and XGB applied to surface ocean pCO2. This is our novel contribution.

*2) There are lots of technical difficulties. The authors stated that Large Ensemble Testbed (LET) consists of 100 members across four initial-condition ensemble models, and each member is a representation of the real ocean climate system. In my understanding, data from these members are actually from the Earth system models. How accurate are these modeled data particularly those (SST, SSS, MLD, Chl-a, .etc) used to train the pCO2 model? How accurate are the pCO2 from the LET comparing to the in situ observations (SOCAT v5)? Without any evaluation, it is questionable to say these modeled data represent the real ocean system. I am not an expert on Earth system models, but why the authors say '100-member LET consists of 25 randomly selected member ...' (L114-115)? What is difference between these members? The authors argued that the use of many members was to test the reconstruction capabilities of the ML across different ocean states, however, what is the impact of ocean state differences on the reconstructed pCO2? Also, there are some technical words that are quite difficult to follow without clear explanation (e.g., full field driver data, unseen data, LET). The ML was trained based on grid data at 1 by 1 degree, what is the impact of real spatial variability within the grid on the uncertainties of the reconstructed pCO2?*

Thank you.

We have added text to the introduction manuscript to clarify that the LET is composed of 4 initial condition large ensemble climate model simulations. These are plausible evolutions of the Earth System from 1982-2016. These are the models used for future climate projection by the IPCC and are thus carefully vetted for many variables, including the ocean carbon cycle (e.g. Long et al. 2013 for CESM).

The goal of this work is to ask the question – if we only have pCO2 as sampled in the real world, how likely is it that we could reconstruct global coverage pCO2? What is the relative skill of one method vs. another? If test data comparisons indicate a certain level of skill, does this represent the skill in extrapolation? Clearly, there are not enough data from the real ocean to evaluate extrapolation. So, we use a proxy – Earth System Models – that represent the physical and biogeochemical processes responsible for

pCO2 evolution in the real world. We add text to make this clear.

We have added to the text additional description of the testbed and its purpose. We have clarified our terminology by replacing "features" with "driver data" and explaining specifically what we mean by "unobserved data".

We also add discussion of shortcomings of the LET, such as an inability to assess the impact of sub-gridscale variability, in the last section of the Discussion.

More generally, we have attempted to improve readability by reducing repetition and adding additional sections to the Discussion.

*3) As to the overall structure of the manuscript, the authors presented details of the three ML methods in both Introduction and Methods. The earth system modeled data and SOCATv5 data are not well described, for example, the data coverage both spatially and temporally, and why they are used. In the ML approaches, again, why these three approaches were selected?*

We have added to the text to clarify that only the spatial pattern of SOCATv5 data is used, not these data themselves.

We have worked to reduce repetition about details, but do feel it is useful to our likely readers if we provide a more general overview of the ML approaches in the introduction, in addition to specific details of our implementation in Methods and Appendix.

In the introduction, we state that the reason these approaches are selected is because they are currently very common for a variety of industry and scientific applications, and because they have been used for pCO2 extrapolation using real data by authors such as Gregor et al. 2019.

Specific comment:

*L244: Statistics to the 'unseen' data is different from those listed in Table 1.*

We have carefully checked to be sure that the MAE statistics cited in the text on line

244-246 are the same as in Table 1. No change to the text is required.

---

## Author Comment (AC2) · 4 Feb 2021

This work by Stamell et al. compares the performance of three machine learning approaches, i.e., feed forward neural network (NN), XGBoost (XGB) and random forest (RF), based on the Large Ensemble Testbed. The authors did a lot of work, however, there are many unclear parts in the manuscript.

Major comments:

*1. The literature review in this manuscript only mentioned previous studies using SOM-FFN to interpolate the pCO2 field. What about the other methods, especially the three methods tested in this study? Have they been used in estimating pCO2 field before? What are the major improvements of this study?*

Thank you for this comment. In the introduction, we also discuss Gregor et al. 2019 in which methods in the class of XGB and RF were applied. Also, in response to the comment from Reviewer 1, we clarify that the goal of this work is to quantify extrapolation uncertainty for NN, RF and XGB applied to surface ocean pCO2.

*2. The SOM-FNN performed well in interpolating the pCO2 field, but is likely to overestimate in the Southern Ocean. Is this issue improved in the three methods from this study?*

Thank you for this comment. In the Conclusion, we have expanded the third paragraph to explicitly address this point.

"Decadal variability is of particular interest to the ocean carbon cycle community (Landschützer et al., 2015; Gruber et al., 2019). We have previously shown that the commonly-used SOM-FFN observation-based pCO2product (Landschützer et al., 2016) likely overestimates the amplitude of Southern Ocean decadal variability due to data sparsity (Gloege et al., 2021). Here, we also find overestimation of the amplitude of decadal variability for all approaches. Nonetheless, we do find that the NN performs slightly better than XGB (Figure 4). The creation of a non-linear mapping, without creating distinct regions in driver data space, appears to lead the NN to better extrapolate to the poorly-sampled decadal timescale."

*3. I am a little confused about the data used to test the three ML methods. What are the target data or ground truth data when training the model? The data from the Large Ensemble Testbed (LET) are the ensemble of Earth system models, which are not observational data. While the SOCATv5 data product, which are actual measurements data, seems not to be included in the model training. Please clarify.*

Thank you for this comment. We now state specifically in the introduction that SO-CATv5 data are not directly used. We use only the pattern of sampling of pCO2 that occurred in SOCATv5. All data are drawn from the LET ensemble members. We have clarified in the text that the goal of this work is to ask the question – if we only have

pCO2 as sampled in the real world, how likely is it that we could reconstruct global coverage pCO2? What is the relative skill of one method vs. another? If test data comparisons indicate a certain level of skill, does this represent the skill in extrapolation? Clearly, there are not enough data from the real ocean to evaluate extrapolation. So, we use a proxy – Earth System Models – that represent the physical and biogeochemical processes responsible for pCO2 evolution in the real world.

We do not attempt to predict real-world pCO2 in this effort. Our goal is to understand the skill of methodologies currently in use to make such predictions. We have endeavored to clarify this throughout.

*4. How are the train (60%), validate (20%) and test data (20%) split? Are they spatial-temporal randomly divided, or according to the locations or times? Different split methods lead to the evaluation of different model abilities. Split according to locations indicates the model's ability in spatial interpolation, while split according to times indicates the model's ability in temporal prediction. Please clarify.*

The split is random in both space and time. We add clarification to the text of section 2.2. and A2 to clarify this.

Minor comments:

*What is the sample size of the data? Line 196: "fianlly" should be "finally"*

Thank you, we have corrected this typo. We have also added the dataset size. It is $\sim 14.2$M total, with $\sim 220$K for training, validation, and testing.